# Classification of Gravity Matching Areas Using PSO-BP Neural Networks based on PCA and Satellite Altimetry Data over the Western Pacific

**DOI:** 10.3390/s22249892

**Published:** 2022-12-15

**Authors:** Jingwen Zong, Shaofeng Bian, Yude Tong, Bing Ji, Houpu Li, Menghan Xi

**Affiliations:** 1Department of Electrical Engineering, Naval University of Engineering, Wuhan 430033, China; 2Key Laboratory of Geological Survey and Evaluation of Ministry of Education, China University of Geosciences (Wuhan), Wuhan 430072, China; 3State Key Laboratory of Geodesy and Earth’s Dynamics, Innovation Academy for Precision Measurement Science and Technology, Chinese Academy of Sciences, Wuhan 430077, China

**Keywords:** gravity matching navigation, optimization classify, PCA theory, underwater navigation, gravity satellite missions

## Abstract

For inertial navigation systems (INS), as one of the major methods for underwater navigation, errors diverge over time. With the development of geophysical navigation technology, gravity navigation has become an effective method of navigation. Significant changes in the gravity characteristic of the matching region ensure that gravity matching navigation works effectively. In this paper, we combine artificial intelligence algorithms and statistical metrics to classify gravity-matching navigation regions. Firstly, this paper analyzes and extracts gravity anomaly data from a matching region in different ways. Then, a particle swarm optimization (PSO) algorithm is used to optimize the network weights of a back propagation (BP) NN. Finally, based on principal component analysis (PCA) theory and PSO-BP NN, this paper proposes the PPBA method to classify the matching area. Moreover, the Terrain Contour Matching (TERCOM) matching algorithm and gravity anomaly data from the Western Pacific are used to verify the classification performance of the PPBA method. The experiments prove that the PPBA method has a high classification accuracy, and the classification results are consistent with the matching navigation experimental results. This work can provide a reference for designing navigation regions and navigation routes for submarines.

## 1. Introduction

INS [1,2] is one of the major methods for underwater navigation. However, INS errors diverge over time. INS cannot provide accurate navigation information over a long period of time; therefore, it must be corrected through other navigation methods to suppress the growth of position errors. Underwater autonomous navigation is the key technology for the realization of effective marine underwater military strategy [3].

Geophysical navigation [4,5] has become effective in underwater autonomous navigation. At present, geophysical navigation can be aided by terrain, gravity, and geomagnetism. However, underwater terrain matching auxiliary navigation needs to transmit sounding signals, which makes it difficult to ensure the concealment of submarines [6,7]. Moreover, the geomagnetic field is a weak magnetic field [8], which makes it susceptible to interference. At the same time, it is difficult to create a high-resolution geomagnetic reference map for a large region. By contrast, the characteristics of the ocean gravity field are stable, and the matching and positioning accuracy is high [9]. The use of Earth’s gravity field to realize the navigation of underwater vehicles is an important research direction for underwater autonomous navigation. 

Gravity matching navigation can work effectively when the gravity characteristics of the matching navigation region change obviously, that is, the region to be matched has good matching characteristics. Therefore, the characteristic parameters of the underwater gravity field are of great significance, as they reflect the characteristics of the gravity field of the matching region. It is urgent to discern the characteristic parameters of the underwater gravity field, select the matching metric criteria, and select the methods of regional matching.

Traditional classification methods have used one of the characteristic gravity parameters such as standard deviation, latitude and longitude roughness, anomaly variation coefficient, kurtosis coefficient, or skew coefficient for analysis and comparison; the disadvantage is that this brings a certain randomness and singleness. Considering the above two shortcomings, the singleness of the parameter and no consideration of directionality, the comprehensive characteristic parameter method has been proposed by various experts and scholars. The analytic hierarchy process was proposed as a new selection criterion of gravity matching [10,11]. Li [12] proposed a new principal component-weighted average normalization method to obtain the overall characteristic parameter, which is used to divide the excellent suitability region, general suitability region, and non-suitability region; Wang studied the influence of these data on a gravity-aided INS from the perspective of a gravity anomaly reference map to evaluate the feasibility and performance of gravity navigation in different navigation regions in advance [13]. Zitova found, in research on scene matching, that a scene region suitable for the matching algorithm must be selected before realizing scene matching [14]. The correlation radius and correlation function are used to estimate the availability of the matching region effectively [15]. Fractal theory (FD) [16,17,18,19] was proposed as a comprehensive characteristic parameter method which considered the directivity of the matching region to analyze a gravity anomaly sequence and assess the gravity field characteristics for the first time. This method proved that the matching region selected by the improved method not only had larger coverage but also better directivity. However, when using a navigation sensor and a gravity anomaly reference map under current technical conditions, how to quantitatively evaluate and classify the gravity matching in different marine environments remains a problem in need of a solution.

Artificial neural networks (ANN) play an important role in the domain of intelligent control, and they have allowed for great progress in many fields, such as neuroscience, mathematics, statistics, and computer science [20,21,22]. BP NN have become one of the most widespread applications due to the fact that the BP NN algorithm is simple and plastic. The trainers in this group are mostly mathematical optimization techniques that aim to find the maximum performance (minimum error). However, there exist two disadvantages: falling into local minima easily and having a slow convergence [23,24]. Local optima refer to the sub-optimal solutions in a search space that might mistakenly be assumed as the global optimum. The challenge here is the unknown number of runs that a trainer requires to be restarted with different initial solutions to find the global optimum. To address these problems, scholars have proposed many solutions, including the Conjugate gradient [25,26], Newton [27], Gauss-Newton [28], and Levenberg-Marquardt methods. In addition, some of the most popular multi-solution trainers in the literature are: Genetic Algorithm (GA) [29,30], PSO [31,32,33], Ant Colony Optimization (ACO) [34], Artificial Bee Colony (ABC) [35,36], and Differential Evolution (DE) [37]. The reason as to why such optimization techniques have been employed as training algorithms is their high performance in terms of approximating the global optimum. At the same time, there are some recent real-life applications of recent ANN. Wang proposed an approach for source localization in underwater ocean waveguides based on machine learning that achieves a good localization performance with high SNRs [38]. Seyedali employed the recently proposed Grey Wolf Optimizer (GWO) for training Multi-Layer Perceptron (MLP), which proved that the GWO algorithm is able to provide very competitive results in terms of improved local optima avoidance and also demonstrates a high level of accuracy in the classification and approximation of the proposed trainer [39]. Abbas used a fuzzy model of control parameters of the whale optimization algorithm which was designed to train an MLP-NN and proposed fuzzy logic to adjust the Chimp optimization algorithm’s control parameters to classify marine mammals. As the simulation results show, Abba’s algorithms can identify the global optimal, avoid local optimization, and have better performance for classifying [40,41]. Wang proposed a new meta-heuristic method called the sine-cosine algorithm (SCA), which was used to train RBF NNs and which provided better results in terms of convergence rate and classification accuracy compared to the standard algorithms [42]. This all shows that ANN has been widely used in classification learning.

PSO is a swarm intelligence optimization method based on the swarm behavior of biota. The algorithm has the advantages of fewer parameters, rapid convergence, and strong global search ability, and it is widely used in pattern recognition, signal processing, intelligent control, and other fields. According to this idea, this paper adopts PSO to optimize a BP NN. 

In this paper, we introduce an efficient PSO algorithm to optimize a BP NN and combine these with the PCA method to propose the PPBA method, which is used to classify and identify the gravity matching region in the entire research area. The PPBA method starts with a large number of characteristic gravity data and ends with the labels “suitable” or “unsuitable” by using the trained PPBA method. The experimental results for the use of TERCOM algorithms show that the navigation errors in good matching regions are much better than those in non-matching regions. The correctness of the PPBA classification results is consistent with the experimental matching navigation results. Thus, the classification evaluates the feasibility and performance of gravity matching navigation in different navigation regions in advance. This provides a reference for designing navigation regions and navigation routes for submarines.

The rest of this paper is organized as follows. In Section 2, the PPBA method is constructed. In Section 3, the experimental materials are introduced. Section 4 presents the experimental results. Section 5 is a discussion, and Section 6 draws conclusions.

## 2. Methods

### 2.1. Gravity Characteristic Selection Method

Significant changes in the gravity characteristic of the matching region ensure that gravity matching navigation works effectively. Better matching navigation results may be gained when gravity changes are much rougher in the matching area. Conversely, smooth gravity changes may lead to large errors and even invalid positioning results. Drawing on the definition of the statistical parameters of terrain features, the standard deviation of gravity anomaly, the standard deviation of slope, roughness, difference entropy of gravity anomaly, FD, average gravity difference, kurtosis coefficient, and relevant coefficient comprise the characteristic parameters of the gravity field.

At present, underwater gravity field data are usually stored in the form of grid data. It is assumed that the gravity data grid of a certain gravity field region is M × N, and the gravity anomaly at Grid(i,j) of the gravity field is G(i,j). In order to analyze the adaptation characteristics of the local gravity field, a local window with a size of m × n is designed to calculate the various statistical parameters of the local gravity field. Finally, the matching characteristics of each local gravity field are combined to obtain the gravity matching characteristics of the whole region. The main characteristic parameters of the gravity field are analyzed and deduced as follows.

1.Standard deviation of gravity anomaly (σ):

(1)$$\begin{array}{c} \sigma =\sqrt{\frac{1}{mn-1}\overset{m}{\underset{i=1}{\sum }}\overset{n}{\underset{j=1}{\sum }}{\left(\left(g\left(i,j\right)-\bar{g}\right)\right)}^{2}}, \end{array}$$
where $\bar{g}=\frac{1}{mn}\sum_{i=1}^{m}\sum_{j=1}^{n},\;g\left(i,j\right)$ is the mean of the gravity anomaly, and σ is the standard deviation of the gravity anomaly, which is used to describe the overall variation characteristics and dispersion degree of the gravity anomaly data. That is, when the gravity field features are obvious, the σ is large. On the contrary, when the σ is small, it means that the fluctuation of the gravity anomaly map is relatively flat.

2.Standard deviation of slope ($\sigma_{S}$):

(2)$$\begin{array}{c} \sigma_{S}=\sqrt{\frac{1}{mn-1}\overset{m}{\underset{i=1}{\sum }}\overset{n}{\underset{j=1}{\sum }}{\left(\left(S\left(i,j\right)-\bar{S}\right)\right)}^{2}}, \end{array}$$
where $\bar{S}=\frac{1}{mn}\sum_{i=1}^{m}\sum_{j=1}^{n}S\left(i,j\right)$, $S_{\emptyset }=\frac{1}{2*△\mathrm{Grid}}\left(g\left(i+1,j\right)+g\left(i+1,j+1\right)-g\left(i,j\right)-g\left(i,j+1\right)\right)$, $S_{\lambda }=\frac{1}{2*△\mathrm{Grid}}\left(g\left(i,j+1\right)+g\left(i+1,j+1\right)-g\left(i,j\right)-g\left(i+1,j\right)\right)$, $S=\arctan\left(\sqrt{S_{\phi }^{2}+S_{\lambda }^{2}}\right)$, $\sigma_{S}$ is the standard deviation of the slope, S is the slope of the gravity field, $S_{\emptyset }$ and $S_{\lambda }$ are the slope of gravity field in the latitude and longitude directions, respectively, ΔGrid is the length of the grid side of the gravity field, and $\bar{S}$ is the mean of the gravity anomaly in the local calculated window. The slope of the gravity field reflects the rate of gravity change in the longitude and latitude directions, and the $\sigma_{S}$ reflects the characteristics of the rate of the slope change. The dispersion of gravity is positively correlated with the $\sigma_{S}$, that is, the faster the slope changes, the more obvious the change in the gravity field.

3.Roughness of gravity anomaly (R):

(3)$$\begin{array}{c} R=\left(\frac{R_{\phi }+R_{\lambda }}{2}\right), \end{array}$$(4)$$\begin{array}{c} R_{\phi }=\frac{1}{mn-1}\overset{m}{\underset{i=1}{\sum }}\overset{n-1}{\underset{j=1}{\sum }}\left|g\left(i,j\right)-g\left(i,j+1\right)\right|, \end{array}$$(5)$$\begin{array}{c} R_{\lambda }=\frac{1}{mn-1}\overset{m-1}{\underset{i=1}{\sum }}\overset{n}{\underset{j=1}{\sum }}\left|g\left(i,j\right)-g\left(i+1,j\right)\right|, \end{array}$$
where R is the roughness of the gravity field [43], and $R_{\emptyset }$ and $R_{\lambda }$ are the slope of the gravity field in the latitude and longitude directions, respectively. The R is used to reflect the smoothness of the trend surface of the gravity field, depicting the fluctuation of the gravity field in this region. The anomalous divergence of gravity is positively correlated with R, that is, the greater the R, the more obvious the change in the gravity field. 

Difference entropy of gravity anomaly (H):(6)$$H=-\overset{m}{\underset{i=1}{\sum }}\overset{n}{\underset{j=1}{\sum }}P_{ij}\log_{2}P_{ij}$$
where $P_{ij}=\frac{D_{ij}}{\sum_{i=1}^{m}\sum_{j=1}^{n}D_{ij}}$, $D_{ij}=\frac{\left|g\left(i,j\right)-\bar{g}\right|}{\bar{g}}$, H is the difference entropy of the gravity anomaly [44], $P_{ij}$ is the probability of difference entropy, and $D_{ij}$ is the difference in the gravity anomaly. Entropy can be used to represent the amount of information contained in an event. To a certain extent, entropy reflects the change in gravity data, and the dispersion of gravity is negatively correlated with entropy, that is, where the change in gravity data is small, the entropy value is large, and where the change is large, the entropy value is small.

4.Relevant coefficient (RC):

(7)$$RR=\left(\frac{RC_{\phi }+RC_{\lambda }}{2}\right)$$(8)$$RC_{\phi }=\frac{1}{m\left(n-1\right)\sigma^{2}}\overset{m}{\underset{i=1}{\sum }}\overset{n-1}{\underset{j=1}{\sum }}\left(g\left(i,j\right)-\bar{g}\right)\left(g\left(i+1,j\right)-\bar{g}\right)$$(9)$$RC_{\lambda }=\frac{1}{n\left(m-1\right)\sigma^{2}}\overset{m-1}{\underset{i=1}{\sum }}\overset{n}{\underset{j=1}{\sum }}\left(g\left(i,j\right)-\bar{g}\right)\left(g\left(i,j+1\right)-\bar{g}\right)$$
where RC is the relevant coefficient of the gravity field, and $RC_{\emptyset }$ and $RC_{\lambda }$ are the relevant coefficients of the gravity field in the latitude and longitude directions, respectively. The RC of the gravity field is used to reflect the degree of the gravity anomaly, where the gravity anomaly is negatively correlated with the relevant coefficient. The larger the relevant coefficient, the higher the degree of relevance between adjacent grid points. At this time, the change of the gravity anomaly in the region is small.

5.Average gravity difference (AG):

(10)$$AG=\frac{1}{2}(\frac{1}{m\left(n-1\right)}\overset{m}{\underset{i=1}{\sum }}\overset{n-1}{\underset{j=1}{\sum }}Gx_{ij}+\frac{1}{n\left(m-1\right)}\overset{m-1}{\underset{i=1}{\sum }}\overset{n}{\underset{j=1}{\sum }}Gy_{ij})$$
where $Gx_{ij}\left(i=1,2,\cdots ,m;j=1,2,\cdots ,n-1\right)$ are the gravity differences between neighboring grids in *x*, and $Gy_{ij}\left(i=1,2,\cdots ,n;j=1,2,\cdots ,m-1\right)$ are the gravity differences between neighboring grids in *y*. The AG quantitively describes the degree of gravity change in a region or regions, which is defined as the AG between neighboring grids in gravity databases or maps [45].

6.Fractal dimension (FD):

(11)$$\log A\left(\varepsilon \right)=C_{1}\log\varepsilon +C_{0},$$
where A(ε) is the surface area of the fractal dimension, ε is the scale to use when measuring $C_{1}=2-D$, which is the slope of the fitted line, and $C_{0}$ is a constant of the fractal dimension. Peleg et al. adopted Mandelbrot’s idea and extended it to surface region calculations [46]. In this extension, the map can be viewed as a gravity field surface whose gravity anomaly from the normal ground is proportional to the image’s gray value. Then, all points at distance ε from the surface on both sides create a blanket of thickness, 2ε. The estimated surface region is the volume of the blanket divided by 2ε. For different values of ε, the blanket region can be calculated by $A\left(\varepsilon \right)=F\varepsilon^{2-D}$. The FD is used to reflect the complex irregularity of the shape of the local gravity field. The dispersion of gravity is positively correlated with the FD, that is, the larger the FD, the more obvious the local gravity field changes are.

7.Kurtosis coefficient (PK):

(12)$$PK=\frac{K_{c1}}{\sigma^{4}}\overset{m}{\underset{i=1}{\sum }}\overset{n}{\underset{j=1}{\sum }}{\left(g\left(i,j\right)-\bar{g}\right)}^{4}-\frac{K_{c2}}{\sigma^{4}}{\left[\overset{m}{\underset{i=1}{\sum }}\overset{n}{\underset{j=1}{\sum }}{\left(g\left(i,j\right)-\bar{g}\right)}^{2}\right]}^{2},$$
where $K_{c1}$ and $K_{c2}$ are the coefficients of PK. The term kurtosis was first coined by statistician; it is a measure of how flat or peaked a distribution of data is. The PK is used to describe the sharpness or flatness of the gravity anomaly distribution, which is achieved by comparison with the PK of the standard normal distribution.

In this paper, the above parameters are used to describe the statistical characteristics of the gravity field from different dimensions to divide the gravity matching regions, as shown in Table 1. However, a single parameter has limitations in describing the characteristics of a gravity field, as it can only reflect the characteristics of the gravity field from a certain aspect. It is completely insufficient and inaccurate to express the matchable of a gravity matching region by any one of the above statistical parameters. Therefore, parameters of multiple dimensions are usually used for comparison and analysis in the process of actually analyzing the matching of the navigation region. At the same time, methods that combine parameters are used to determine whether a navigation region is good for matching.

### 2.2. The Proposed Principal-Component-Analysis-Based Method

In situations where a wide variety of gravity field statistical characteristic parameters reflect different aspects of gravity field information, how to ensure that the loss of the original gravity field information is limited to within a small range, and how to establish the selection criteria for the matching region by comprehensively using the above different gravity field characteristic parameters have become key issues. 

The PCA method is one of the most widely-used data dimensionality reduction algorithms, which is used to analyze the inherent structure of data. The method helps to reduce data dimensionality by rotating coordinate axes. The proposed PCA involved an eigenvalue decomposition to produce eigenvalues and eigenvectors for representing the variation of the origin data. A set of correlated high-dimensional sensed data were transformed into a set of uncorrelated lower-dimensional data. In order to compare and analyze the size and features of the characteristic parameters, the normalization method was used to standardize the origin data to eliminate the influence of data dimensionality. This method uses calculations based on the mean and standard deviation of the original data. The original data that are processed conform to a standard normal distribution, that is, the mean is 0, and the standard deviation is 1. The steps of the PCA method can be summarized by the following:(1)Start;(2)Step (1) Select characteristic parameter $X_{1},X_{2},X_{3},\cdots ,X_{k}$;(3)Step (2) Standardize X and compute its covariance matrix;(4)Step (3) Calculate the eigenvalues and eigenvectors of the covariance matrix;(5)Step (4) Calculate the PCA expression and the new principal component index;(6)End.

### 2.3. The PSO-BP Combined Artificial Neural Network Algorithm

#### 2.3.1. BP NN

Since Rulmhart and the parallel distributed processing group put forward the BP algorithm in 1986 [47], ANN researchers began to pay attention to BP NN. BP networks can learn and store a large number of input-output pattern mapping relationships without revealing the mathematical equations that describe these mapping relationships in advance. Analogous to the least squares linear regression problem, when solving a data fitting line, a method is used to make the “deviation” between the predicted value and the actual value as small as possible. The BP NN updates the parameters of the network once through one forward propagation and one back propagation. The first layer is always called the input layer, whereas the last layer is called the output layer. Other layers between the input and output layers are called hidden layers. The process of backpropagation uses the principle of the gradient descent method to continuously adjust the weights and thresholds of the network to minimize the sum of the squared errors of the network and, finally, to make the network approach the real relationship. However, the flaw of the gradient descent method is that, if the location of the initial point is not selected properly, the local optimal solution is found instead of the global optimal solution. At the same time, slow learning efficiency and slow convergence are also disadvantages of BP networks.

#### 2.3.2. PSO

The PSO algorithm was put forward by Kennedy and Eberhar in the United States in 1995 [48], which is a kind of evolutionary algorithm based on intelligence, and which aims to simulate the unpredictable movement of birds. The main idea is to regard a solution for the problem as a particle. The particle’s process of searching for its optimal solution refers to the process of searching for food. Similar to a GA, the PSO algorithm is an iterative optimization algorithm, but without operations such as crossover and mutation. The search process is carried out by particles continuously updating their speed and position in the solution space to follow the optimal particle.

The PSO algorithm can be used to optimize the network weights of a BP NN. A traditional BP NN uses error BP to adjust the network connection weights. This method easily falls into the local optimal solution, while the PSO algorithm can search over a larger space to avoid the above problems to a certain degree. The above process is described in Figure 1.

Among them, Pbest represents the d-th dimensional components of the best position, which refers to the *i*-th individual particle. Pgd represents the d-th dimensional components of the optimal value Gbest in the particle swarm; Vid(t) represents the *d*-th dimensional components of particle i’s speed after iterating for a *t*-th time; Xid(t) represents the *d*-th dimensional components of particle i’s position after the *t*-th time. The PSO algorithm has the advantages of a smaller calculation and rapid convergence in dealing with high-dimensional functions.

### 2.4. The PCA-PSO-BP Combined Algorithm Method

Thus far, we have analyzed the advantages and the disadvantages of BP NN and the PSO algorithm. Data from these networks are typically multidimensional. To address this problem, this paper combines a PSO-BP NN with the PCA method, which can effectively reduce data dimensionality. Using the gravity field characteristic parameters and matchable labels as input, the mapping relationship between the two is obtained by the PSO-BP NN. Firstly, the gravity field characteristic parameters of the gravity matching region are used as input. Secondly, the corresponding matchable labels are obtained by the PSO-BP NN, which is trained for classification. Finally, the prediction selection of the gravity matching region is completed. The steps of the PPBA method are as follows:

Step 1: PCA preprocess the original data. The gravity field characteristic parameters σ, $\sigma_{S}$, R, H, RR, AG, FD, and PK are used as the original data of PCA, which are defined as $X_{1},\;X_{2},\;X_{3},\;\cdots ,\;X_{k}$. Firstly, the original data are normalized to obtain the standard matrix Y. Matrix Y consists of the characteristic parameters and can be expressed as:(13)$$Y=\left[\begin{array}{ccccc} y_{1} & y_{2} & y_{3} & \cdots & y_{k} \end{array}\right]=\left[\begin{array}{ccccc} y_{11} & y_{12} & y_{13} & \cdots & y_{1k}\\ y_{21} & y_{22} & y_{23} & \cdots & y_{2k}\\ y_{31} & y_{32} & y_{33} & \cdots & y_{3k}\\ \vdots & \vdots & \vdots & \ddots & \vdots \\ y_{n1} & y_{n2} & y_{n3} & \cdots & y_{nk} \end{array}\right],$$
where $y_{i}=\frac{x_{i}-\bar{x}}{s}$, $\bar{x}=\frac{1}{n}\sum_{i=1}^{n}x_{i}$, $s=\sqrt{\frac{1}{n-1}\sum_{i=1}^{n}{\left(x_{i}-\bar{x}\right)}^{2}}$, $y_{i}$ is the *i-*th normalized sample column vector, $y_{ij}$ is normalized with k raw samples (rows) on n process variables (columns), $x_{i}$ is the *i-*th original data, $x_{i}\in X_{i}$, $\bar{x}$ is the mean of the original data, and s is the standard deviation of the original data. Then, covariance matrix R demonstrates the correlation of the standard matrix, which can be expressed as:(14)$$R={\left[r_{ij}\right]}_{k\times k}=\frac{Y^{T}Y}{n-1}$$

After the eigenvalue decomposition of R, that is, $\left|R-\lambda I_{k}\right|=0$, k eigenvectors $\lambda_{i}$ can be calculated (i is 1, 2, ⋯, p, ⋯, k). The rate of information contribution can be calculated as follows:(15)$$b_{i}=\frac{\lambda_{i}}{\sum_{i=1}^{k}\lambda_{i}},\;i=1,2,\cdots ,p,\cdots ,k,$$
where $b_{i}$ is the rate of information contribution of the *i-*th principal components. Usually, p(p≤k) principal components are taken to replace the m original variables so that the cumulative contribution rate of the principal components is greater than 95%. Finally, through linear combination, k normalized gravity field characteristic parameters are recombined to generate p new parameters, $Z_{1},\;Z_{2}$, $\cdots ,\;Z_{p}$, which are not related to each other. The linear combination can be expressed as:(16)$$\left\{ \begin{array}{c} Z_{1}=a_{11}\lambda_{1}^{-\frac{1}{2}}y_{1}+a_{12}\lambda_{1}^{-\frac{1}{2}}y_{2}+\cdots +a_{1k}\lambda_{1}^{-\frac{1}{2}}y_{k}\\ Z_{2}=a_{21}\lambda_{2}^{-\frac{1}{2}}y_{1}+a_{22}\lambda_{2}^{-\frac{1}{2}}y_{2}+\cdots +a_{2k}\lambda_{2}^{-\frac{1}{2}}y_{k}\\ \vdots \\ Z_{p}=a_{p1}\lambda_{p}^{-\frac{1}{2}}y_{1}+a_{p2}\lambda_{p}^{-\frac{1}{2}}y_{2}+\cdots +a_{pk}\lambda_{p}^{-\frac{1}{2}}y_{k} \end{array},\right.$$
where $a_{ij}$ is the loading of the *j-*th eigen parameter on the *i-*th principal components. After processing via the PCA method, the new gravity field characteristic parameters are sorted according to the contribution rate, where the larger the contribution rate, the more important the principal component. This paper selects the first k principal components $Z_{1},\;Z_{2},\cdots ,\;Z_{k}$ as input for the NN.

Step 2: using a gravity matching algorithm to calculate matching probability (MP). In this paper, the TERCOM algorithm is selected to carry out the gravity matching navigation experiment, which is used to calculate the MP as the index to measure the matchable. The method divides the experimental sea into regions of the same size and conducts a large number of underwater gravity matching navigation experiments to calculate the MP. Based on a large number of simulation experiments and empirical analyses, it is determined that $P_{suit}$ is the MP ≥ 85% in the matching area. According to the accuracy requirements of gravity matching navigation, we set labels with MP $>{\;P}_{suit}$ as “suitable” and MP $\leq {\;P}_{suit}$ as “unsuitable.” This paper will select the MP as an input for the BP NN.

Step 3: BP NN initializing. The new parameters $Z_{1},\;Z_{2},\;\cdots ,\;Z_{p}$ are set as the input layer of the BP NN, as determined by λ, which is the matrix’s dimensionality. The number N serves as the output layer, as determined by the output properties. The number Q serves as a hidden layer; however, it has no exact mathematical theorem proving what value to take. On the basis of the empirical function $Q=\sqrt{M+N}+a,\;a=1\sim 10$, we combine the classification evaluation indicators to determine the number of layers of the hidden function. 

Step 4: the PSO algorithm optimizes the BP NN. Each input of the NN is seen as a particle of PSO. We initialize the speed matrix and the position matrix of each particle. The position matrix will be given a random number between 0 and 1. Then, we calculate the fitness of each particle using the following formula:(17)$$\mathrm{FITNESS}=\frac{1}{2}\overset{K}{\underset{i=1}{\sum }}{\left(Yreal-Yi\right)}^{2}$$
where *K* is the number of training samples, Yreal is the true value of the first i samples, and Yi is the predictive value of the first i samples. The optimal solution is the position of minimum fitness for the particle.

Step 5: compare the fitness of each particle. We obtain each particle’s extreme value and the global optimal value.
If Present<Pbest, Present=Pbest, Pbest=x;else Pbest the same
If Present<Gbest, Present=Gbest, Gbest=x;else Pbest the same

Among them, Present represents the current fitness of the particle, Pbest represents the best position of the current particle itself, and Gbest is the best position found by all particles in the current whole population.

Step 6: update the speed and position of each particle. After each iteration, the particle’s speed ($V_{id}$) and the $X_{id}$ will be updated by the individual extreme value and the globe extreme value. Particles update their speed and position through the following formulas:(18)$$V_{id}\left(k+1\right)=\omega V_{id}\left(k\right)+c_{1}r_{1}\left(P_{id}\left(k\right)-X_{id}\left(k\right)\right)+c_{2}r_{2}\left(P_{gd}\left(k\right)-X_{id}\left(k\right)\right)$$
(19)$$X_{id}\left(k+1\right)=X_{id}\left(k\right)+V_{id}\left(k+1\right),$$
where ω is the inertia weight using the linearly decreasing weight strategy, $c_{1}$ and $c_{2}$ are acceleration constants, and $r_{1}$ and $r_{2}$ are uniform random numbers in a range from 0 to 1.

Step 7: update the individual extreme value and the global extreme value. Comparing the individual moderate value and the globe moderate value before iteration obtains the best representation of the individual extreme value.

Step 8: stopping iteration. Through the termination condition of the algorithm, we judge whether the algorithm should stop or not. If the number of iterations reaches the maximum or an expectant error, then we return to Step 8, or else we go back to Step 4.

Step 9: use the eigenvalues in Step 1 and the adaptability labels in Step 2 as input parameters to train a reliable PSO-BP NN.

Step 10: divide the experimental region into multiple sub-regions of the same size, repeat Step 1 to obtain several independent eigenvalues, and input them into the trained PSO-BP NN to complete the classification and identification of the gravity matching region. 

Figure 2 depicts a flowchart of the PPBA classification system for the classification and identification of the gravity matching region.

## 3. Materials

As satellite altimetry and satellite gravity technologies continue to advance, the resolution of the marine gravity field has reached $1^{\prime }\times 1^{\prime }$ based on data processing and further refinements. For example, the Scripps Institution of Oceanography and the University of California San Diego released the $1^{\prime }\times 1^{\prime }$ global marine gravity anomaly model grav.img.24.1, with an accuracy of 3–8 mGal [31,48,49]. To show and implement the PPBA classification method in a straightforward manner, we chose a publicly available collection of sea gravity anomalies as our experimental datasets [50]. This paper sourced the gravity anomaly data from the University of California San Diego website (http://topex.ucsd.edu/, accessed on 4 August 2021), with an original resolution of $1^{\prime }\times 1^{\prime }$ global marine gravity anomaly model grav.img.29.1. The research area for this PPBA experiment that was used for gravity matching navigation was the Western Pacific region, spanning a longitude of 129° E–135° E and a latitude of 24° N–30° N. Figure 3 is a schematic view of the gravity anomaly reference map in the Western Pacific region (the satellite imagery in Figure 3 is from the General bathymetric Chart of the Oceans (GEBCO), https://www.gebco.net/, accessed on 23 June 2022). Currently, the measurement accuracy of the latest LaCoste & Romberg-type and KSS31-type marine gravimeters reaches ±1.0 mGal under relatively poor sea conditions. Taking into consideration the influences of individual errors (i.e., zero drift correction, Eotvos correction, inaccurate location determination, and spatial correction) in marine measurements, an accuracy of ±1.0 mGal−±2.0 mGal is expected [51]. These allow for the use of the gravimeter and marine gravity reference map in underwater aided navigation.

The bilinear interpolation method [52] was employed to transform the gravity anomaly data into 180 m×180 m grid-resolution gravity data, as shown in Figure 4 (the gravity anomaly data in Figure 4 are from the University of California San Diego website (http://topex.ucsd.edu/, accessed on 4 August 2021). Figure 4 shows the gravity anomaly reference map in the test region with 180 m×180 m resolution in two dimensions (4a) and three dimensions (4b), respectively. Statistically, the maximum value of the gravity anomaly in this region is +152.8 mGal, while the minimum value is −168.3 mGal, and the average value is −0.6055 mGal. The experimental region was divided into 400 samples; the range of each sample was $15\times 15\;{\mathrm{Grid}}^{2}$, with a size of approximately 34 km×34 km. All tests were run with Windows 10 (21H2 Version), an Intel(R) Core (TM) i5-7200U CPU @ 2.50 GHz 2.71 GHz, and Matlab2021a.

## 4. Experiments and Results

In order to test the effectiveness and superiority of the PPBA method in the classification and identification of the gravity matching region, we used various characteristic parameters to qualitatively illustrate the relationship between the characteristics of the marine gravity anomaly reference map and the location accuracy of gravity matching, since different parameters may characterize properties with variable emphases. We used Formulas (1)–(12) to analyze the characteristic parameters of the 400 samples (introduced in Section 3) of the experimental region. Because of the influence of unit and dimension, we normalized the feature data using a regularization method to compare and analyze the size and characteristics of the data. 

### 4.1. Characteristic Parameters of the Gravity Field in the Experiment Region

Figure 5 shows a three-dimensional map and the histogram probability statistics of each characteristic parameter, standardized using 400 samples. Figure 5a1,a2–h1,h2 shows a three-dimensional map and the histogram probability statistics of the σ, $\sigma_{S}$, R, H, FD, RC, AG, and PK, respectively. According to the correlation with gravity anomaly shown in Figure 5, it can be seen that the larger the σ, the larger the $\sigma_{S}$, the larger the R, the larger the FD, the smaller the H, the larger the AG, the smaller the RC, the larger the PK, and the richer the characteristic information contained in the gravity field, which is a benefit for gravity matching.

In summary, due to the limited amount of information contained in a single feature quantity, a single gravity field characteristic parameter cannot effectively evaluate matching. We used the PCA method to recalculate and analyze the eight characteristic parameters, using the specific process described in Section 2.2. Among them, the eigenvalues, variance contribution rate, and cumulative contribution rate statistics are given in Table 2.

As shown in Table 2, the cumulative contribution rate of the first five principal components reaches 95%, which meets the selection criteria for principal components. The digital data with eight components was reduced to five components. We take the first five principal components as the following research objects. At the same time, we compute the load number of each characteristic parameter for the first five principal components, as shown in Table 3.

As shown in Table 3, the σ, $\sigma_{S}$, R, FD, and RC occupied the load number with a larger absolute number in the first principal component. At the same time, the H and the AG occupied the load number with a larger absolute number in the second and fourth principal components. In the third principal component, the PK occupied the load number with a larger absolute number. In the fifth principal component, the RC occupied the load number with a larger absolute number. Through the eigenvalues in Table 2 and the load number in Table 3, the principal component linear equation can be calculated as:
(20)$$\left\{ \begin{array}{l} Z_{1}=0.231y_{1}-0.017y_{2}-0.026y_{3}-0.071y_{4}+0.132y_{5}-0.152y_{6}-0.192y_{7}+0.329y_{8}\\ Z_{2}=0.364y_{1}+0.095y_{2}+0.064y_{3}+0.036y_{4}+0.329y_{5}-0.241y_{6}-0.597y_{7}+0.090y_{8}\\ Z_{3}=0.464y_{1}-0.015y_{2}+0.058y_{3}-0.072y_{4}+0.336y_{5}+0.185y_{6}-0.445y_{7}-0.647y_{8}\\ Z_{4}=-0.164y_{1}+0.952y_{2}+0.134y_{3}+0.908y_{4}+0.272y_{5}-0.030y_{6}-0.273y_{7}+0.09y_{8}\\ Z_{5}=-0.591y_{1}-0.116y_{2}-0.341y_{3}-0.167y_{4}+1.003y_{5}-0.752y_{6}+0.219y_{7}-0.278y_{8} \end{array}\right.$$

Figure 6a shows a three-dimensional map of the new principal component-weighted average normalization method [12] in the experimental region. Figure 6b shows a statistical histogram of the PCA method in the experimental region. Li [12] normalized the principal component linear combination matrix to one characteristic parameter, which easily and efficiently classifies the matching region. According to Figure 6a, the gravity characteristics of the northwest and southeast regions chang significantly. According to Figure 6b, characteristic parameters greater than 0.3 accounted for approximately 10%, characteristic parameters greater than 0 accounted for 50%, and values between −0.3 and +0.2 accounted for approximately 88%. According to Figure 6, it can be predicted that regions with characteristic parameters of the experimental region greater than 0.1 have good matchability, and the northwest and southeast parts of the experimental region are more suitable for matching navigation.

### 4.2. Results of the PPBA Method and the Classification Evaluation Index

The data in Equation (20) were processed by the PCA are as input data of the NN, and a PPBA method based on the PSO-optimized BP NN was proposed to perform the classification and prediction of the matching region. To test the effectiveness and superiority of the PPBA method, four types of tests were used. Tset1 is a no-feature analysis of the gravity field and a no-parameter optimization for the BP NN. Tset2 uses the PCA for the gravity field and a no-parameter optimization for the BP NN. Tset3 is a no-feature analysis of the gravity field and uses the PSO-BP NN. Tset4 uses the PCA for the gravity field and PSO-BP NN. We used a confusion matrix and average classification accuracy (AA) as the classification evaluation index. The confusion matrix is shown in Table 4. From the confusion matrix [53], more advanced classification indicators can be obtained: Accuracy, Precision-T, Precision-F, Specificity, and Sensitivity. This paper selects three of these and redefines them. The Accuracy (A), Precision-T (PT), Precision-F (PF), and average classification accuracy (AA) are as follows:Accuracy (A):(21)$$A=\frac{TP+FP}{TP+FP+TN+FN}$$Precision-T (PT):(22)$$PT=\frac{TP}{TP+FP}$$Precision-F (PF):(23)$$PF=\frac{TN}{TN+FN}$$Average classification accuracy (AA):(24)$$AA=\frac{1}{2}\left(\frac{TP}{TP+FP}+\frac{TN}{TN+FN}\right).$$

According to the four test schemes and classification evaluation indicators, the test was carried out, and the statistical test results are shown in the Table 5.

From the above tests, it was shown that the PPBA method we proposed is effective. Therefore, we randomly selected four candidate regions from among the 400 samples as Region-A, Region-B, Region-C, and Region-D for gravity matching navigation simulation analysis and recorded the MP. At the same time, we used the PPBA method to classify the four regions using the labels of “suitable” and “unsuitable.” The results of the simulation and classification are shown in Table 6.

As shown in Table 6, the results of matching probabilities in Region-A were 92.25% and, in Region-B, were 89.50%, and both of them were greater than 85%, which we set as the probability of success for matching navigation. The results of classification by PPBA for both Region-A and Region-B were “suitable,” which is in good agreement with the result of matching probabilities. At the same time, the results of matching probabilities for both Region-C and Region-D were less than 85%, which is in good agreement with the "unsuitable" given by the classification result of PPBA. The above analysis shows that the PPBA method shows promise.

### 4.3. Classification Results by PPBA

To further verify the correctness of the classification results in the four regions, we used the TERCOM algorithm with the continuous optimizing method to conduct INS simulation experiments. The motion mode was uniform linear motion. The simulation parameters were set as follows: an accelerometer of $0.01^{{}^{\circ}}/h$ with a gyro of $10^{-3}\;m/s^{2}$, a sailing speed of $10\;n\;\mathrm{mile}\cdot h^{-1}$, a sailing direction of northeast at $70^{{}^{\circ}}$, an initial position error of 0 m, a velocity error of $0.04\;m\cdot s^{-1},$ and a sailing direction error of 0.05°. In addition, the real-time measured data of the gravimeter were the sum of the sampled value in the gravity anomaly reference map and random noise with a standard deviation of 1 mGal, while the number of sampling points and the sampling period were set as 110 and 2 minutes, respectively. The total sailing time was 3.6 hours. The matching results of the inertial navigation simulation experiment are shown in Figure 7, and the error statistics are shown in the Table 7.

Figure 7a–d shows the matching results of the INS simulation experiment in Region-A, Region-B, Region-C, and Region-D, respectively. Figure 7a–b shows that the real trajectory (red line) and the matching trajectory (yellow line) had a good match in regions with significant changes in gravity anomalies. At the same time, Figure 7c–d shows that the real trajectory (red line) and the matching trajectory (yellow line) had a large deviation in regions with smooth changes in gravity anomalies.

According to Table 7, the latitudinal, longitudinal, and radial errors in regions A and B were much smaller than in regions C and D. Therefore, it is further verified that, compared with regions C and D, regions A and B had good matching performance. Furthermore, the results are consistent with the classification of PPBA. Finally, when the gravity anomaly data of the entire experimental region were input into the PPBA system, we obtained the matching classification map of the experimental region. The gravity matching navigation matchable region in the Western Pacific is shown in Figure 8.

## 5. Discussion

As seen in Figure 8, the white portion is the suitability region of the matching region, and the black portion is the unsuitability region. The suitable regions are mainly concentrated in the northwest and southeast, which is consistent with our previous prediction. This provides a reference for planning navigation regions and designing navigation routes for underwater submarines.

The gravity characteristics of the matching area change significantly, which causes the gravity matching navigation to work effectively. We proposed the PPBA method to extract the gravity characteristics of the matching area for analysis and then input them into the NN to realize the classification of the area. In order to test the effectiveness and superiority of the PPBA method in the classification and prediction of the matching region, we designed four tests. As seen in Table 5, A and AA of test1 were the lowest at 87.00% and 87.60%, respectively. Compared with test1, test2 and test3 performed better in terms of A and AA, where A increased by 4.25% and 6.50%, and AA increased by 3.14% and 4.38%, respectively. Therefore, using PCA for the gravity anomaly and the BP NN using PSO to optimize the parameter was shown to be effective. At the same time, test4, using the PPBA method, yielded the best performance, where A was 96.75%, AA was 96.22%, PT was 92.86%, and PF was 99.57%. According to other studies [45], many traditional methods use one of the characteristic gravity parameters to analyze and compare the matching areas; the disadvantage is that using one parameter brings a certain randomness and singleness. Simultaneously, too many characteristic parameters can lead to information redundancy. Most scholars conduct data processing from the perspectives of information theory and statistics, such as in PCA and the analytic hierarchy process. We chose PCA to preprocess the data, with the purpose of reducing and recombining the gravitational anomaly data in different aspects to prevent insufficient or excess data. Moreover, the preprocessing of the data also assists in determining the correct rate of NN classification, as shown in Table 5. Comparing test1 with test2, it can be seen that A increased by 4.25% and AA increased by 3.14% in the latter. Comparing test3 with test4, it can be seen that A increased by 3.25% and AA increased by 3.79% in the latter. On the other hand, the introduction of artificial intelligence classifiers such as support vector machines [54] and BP NN can reduce the factors of human interference and achieve self-organizing classification. This paper used PSO-BP to help determine the correct rate of NN classification, as shown in Table 5. Comparing test2 with test4, it can be seen that A increased by 5.50% and AA increased by 5.48% in the latter. This paper combined PCA with PSO-BP NN to improve the classification accuracy of the PPBA method via data preprocessing and network optimization. The final classification results are improved, which means that the PPBA method is useful.

According to the classification results, this paper chose four regions for gravity matching navigation simulation analysis and recorded the MP using experiments: two “suitable” regions (Region-A and Region-B) and two “unsuitable” regions (Region-C and Region-D). As seen in Table 6, the MP in Region-A was 92.25% and the MP in Region-B was 89.50%, which are greater than 85%. At the same time, according to the classification results, the Region-A and Region-B were suitable for gravity matching navigation. In addition, the MP in Region-C was 47.06% and the MP in Region-D was 65.31%, which are less than 85%. Meanwhile, according to the classification results, the Region-C and Region-D were unsuitable for gravity matching navigation. Compared with the matching probabilities and Category labels in Table 6, the results of classification and matching probabilities were in good agreement that the PPBA method is available.

Next, we used the TERCOM algorithm with the continuous optimizing method to conduct INS simulation experiments in the four regions (Region-A, Region-B, Region-C, and Region-D). The experimental gravity matching results in the different areas are shown in Figure 7 and Table 7. We can clearly see that the adaptability of the regions had a significant impact on gravity matching navigation. In the “suitable” region, the real trajectory and the matching trajectory match well. As seen in Table 7, the mean errors in the latitude, longitude, and radial directions of Region-A were 0.0881, 0.0607, and 0.0566 n miles, respectively, and the matching accuracy was within 0.1 n miles. Similarly, the mean errors in the latitude, longitude, and radial directions of Region-B were 0.0932, 0.0209, and 0.097 n miles, respectively, and the matching accuracy was within 0.1 n mile. Overall, the “suitable” regions performed well for gravity matching navigation. In contrast, in the “unsuitable” regions, the real trajectory and the matching trajectory did not match well. The maximum error of Region-C and Region-D in the radial direction was 13.217 and 6.4319 n miles, respectively, which indicates a matching navigation failure. The mean errors in the latitude, longitude, and radial directions of Region-C were approximately 4.93, 4.44, and 6.89 n miles, respectively, and the mean errors in the latitude, longitude, and radial directions of Region-D were approximately 1.0758, 2.6116, and 2.8950 n miles, respectively. The matching trajectory drifted in the “unsuitable” regions. The experiments proved that the results shown in Figure 8 and Table 7 are consistent with the classification results. 

Many experts and scholars have proposed different methods to study the suitability of matching regions. The original data are still gravity anomaly data; however, the characteristic parameters selected by each method are different, and, therefore, there is difficulty in creating a unified standard. At present, most papers use statistical methods to analyze and summarize the matching information. Therefore, how to quantitatively evaluate the location accuracy of gravity matching in different marine gravity anomaly characteristics remains a difficult problem to solve. Future works should determine how to make the matching area classification more accurate and faster, so as to provide a larger range and more accurate navigation background maps for gravity matching navigation. It is hoped that in future research, an analytical method can be derived to divide the matching region.

## 6. Conclusions

In this paper, we analyze and extract the gravity anomaly data of a matching area in different ways. Then, based on PCA theory and PSO-BP NN, we propose the PPBA method. The experimental datasets are from the gravity anomaly data on the University of California San Diego website, with an original resolution of 1′ × 1′. The correctness of the PPBA classification results is consistent with the matching navigation experimental results. 

Our method can predict the matchable of matching regions using gravity anomaly characteristics, and experimental verification shows that the prediction results are reliable. Compared with the traditional methods that use one of the gravity characteristic parameters to analyze and compare the matching regions, the PPBA method reduces dimension and regroups and analyzes the data from many aspects, which could better balance local characteristics with the overall characteristics. Compared with the classic BP NN method, the PSO-BP NN searches over a larger space to avoid falling into the local optimal solution and improve the classification accuracy. Further, we implement the automatic identification and classification of gravity matching navigation in experimental regions. Finally, the PPBA method provides a reference for designing navigation regions and navigation routes for submarines.

This paper used the PPBA method to prove that the artificial intelligence classifier can be applied to gravity matching navigation adaptation area classification. However, there are still some shortcomings. In the future studies, we will continue in-depth research in this direction as follows: (1) optimize the preprocessing method of original measurement data and design a more reasonable classification data collection. Improve the calculation method of gravity matching navigation MP; (2) enhance research in ANN. Using the traditional classical methods such as GA, ACO, and ABC as the artificial intelligence classifier into gravity matching navigation adaptation area classification; (3) optimize our classification method by combining the recent deep learning and traditional optimization networks as a comparison to verify the classification results; (4) design of optimal NN structure for gravity matching navigation adaptation area classification.

## Figures and Tables

**Figure 1 sensors-22-09892-f001:**
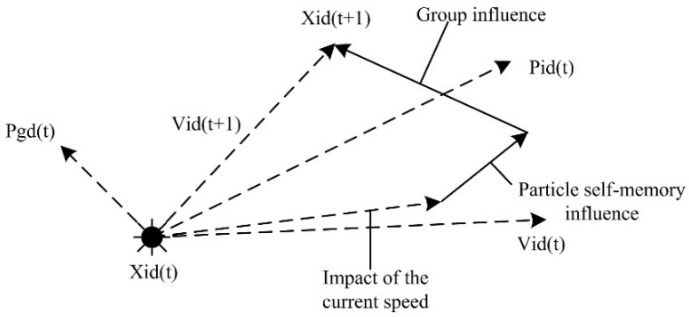
Particle position update of PSO algorithm.

**Figure 2 sensors-22-09892-f002:**
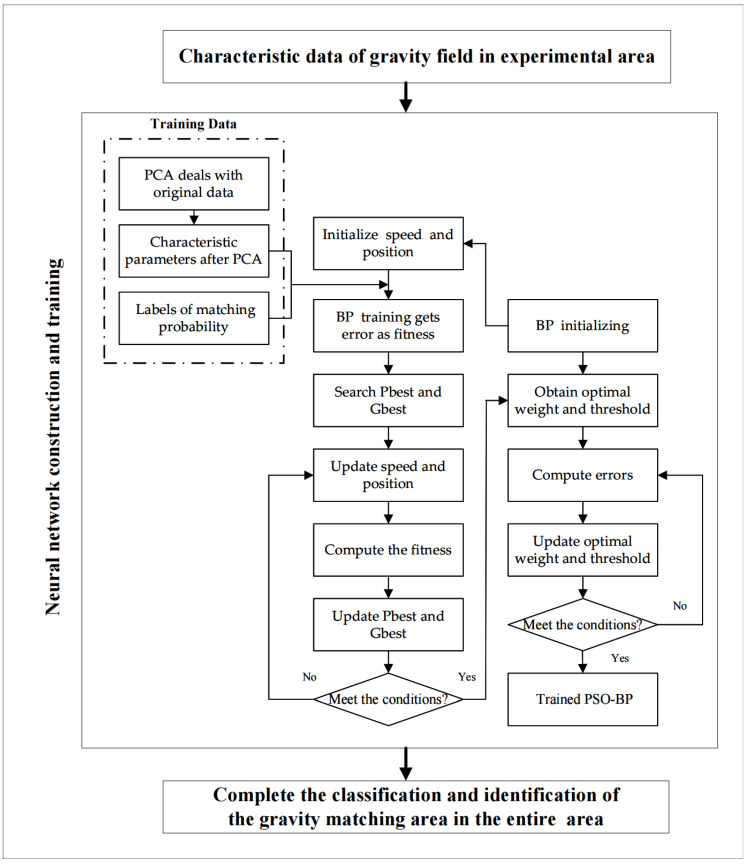
Flowchart of the PPBA classification system.

**Figure 3 sensors-22-09892-f003:**
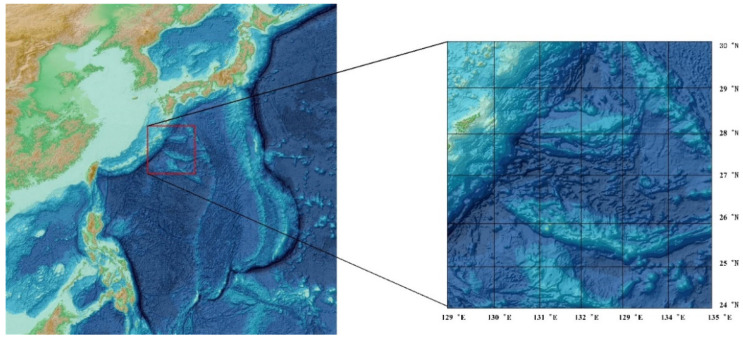
Schematic view of the reference map in the China Western Pacific region (https://www.gebco.net/, accessed on 23 June 2022).

**Figure 4 sensors-22-09892-f004:**
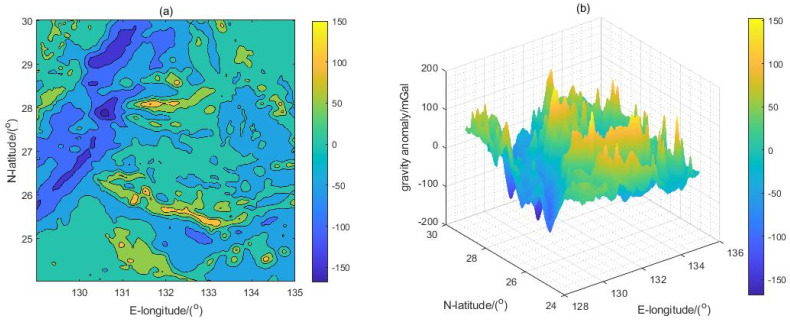
The gravity anomaly reference map in the test region with 180 m×180 m resolution. (**a**) Two-dimensional map; (**b**) three-dimensional map.

**Figure 5 sensors-22-09892-f005:**
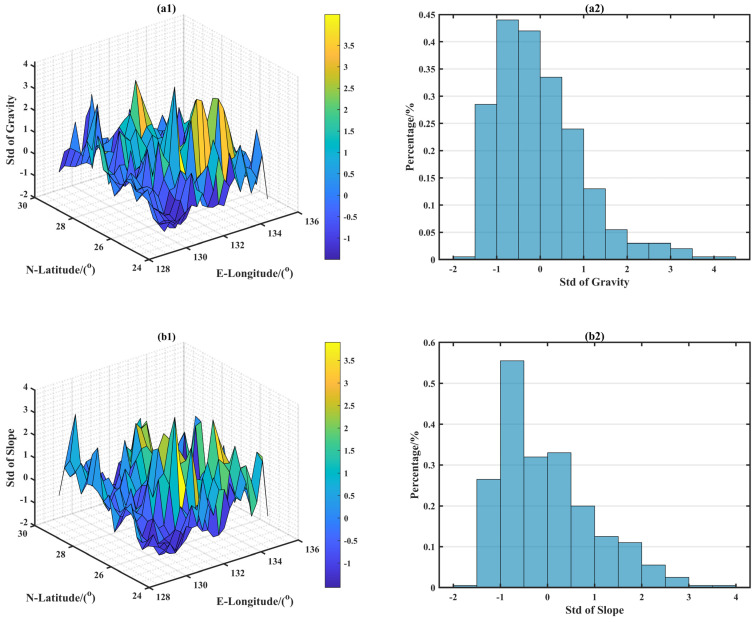
Characteristic parameters map in the experiment region. (**a**) Standard deviation of gravity anomaly; (**b**) standard deviation of slope; (**c**) roughness; (**d**) difference entropy of gravity anomaly; (**e**) fractal dimension; (**f**) relevant coefficient; (**g**) average gravity difference; (**h**) kurtosis coefficient.

**Figure 6 sensors-22-09892-f006:**
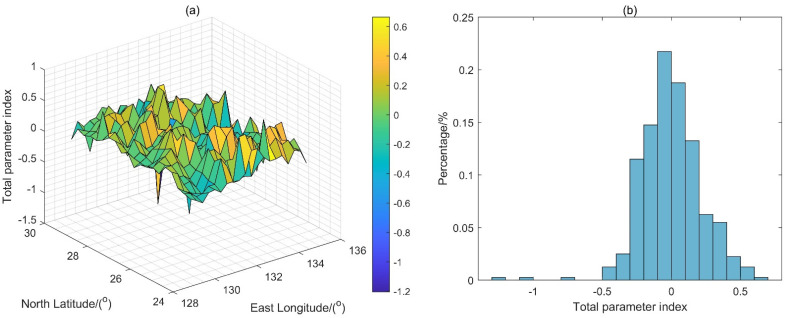
The characteristic parameters map of the experiment region: (**a**) the three-dimensional map of a new principal component-weighted average normalization method [12]; (**b**) the statistical histogram of the PCA.

**Figure 7 sensors-22-09892-f007:**
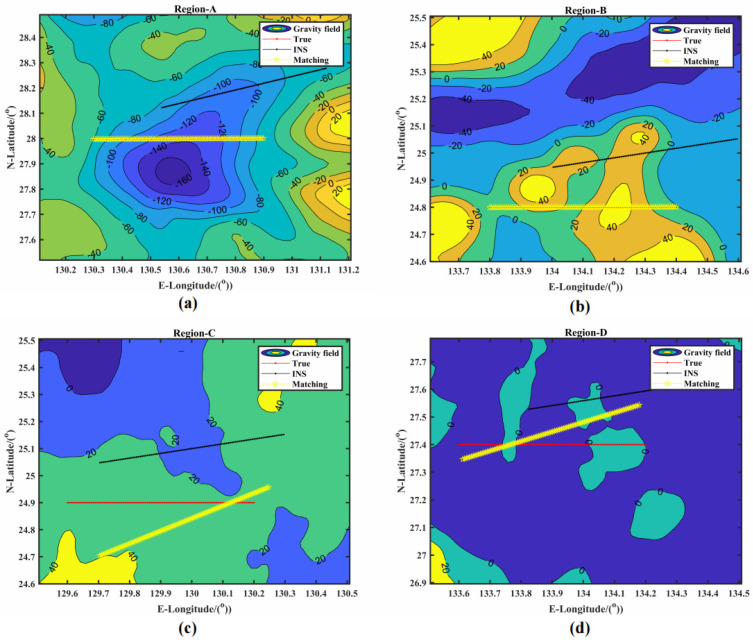
The matching results of the inertial navigation simulation experiment: (**a**) Region-A; (**b**) Region-b; (**c**) Region-C; (**d**) Region-D.

**Figure 8 sensors-22-09892-f008:**
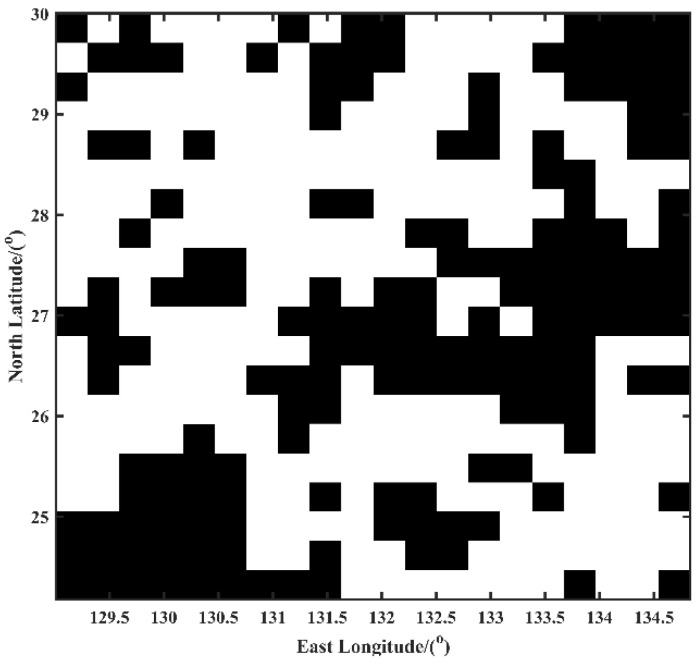
The classification of the Gravity Suitability Region.

**Table 1 sensors-22-09892-t001:** The main characteristic parameters of the gravity field.

Characteristic Parameters of the Gravity Field	Feature Description	Correlation with Gravity Anomaly
σ	overall variation characteristics of the gravity field	Positive
$\sigma_{S}$	characteristics of the rate of slope change	Positive
R	smoothness of the trend surface of the gravity field	Positive
H	the amount of information in gravity field	Negative
RC	relevant between neighboring grid in gravity field	Negative
AG	degree of the gravity changes in regions or regions	Positive
FD	complex irregularity shape of the local gravity field	Positive
PK	sharpness and flatness of the gravity anomaly distribution	Positive

**Table 2 sensors-22-09892-t002:** Statistics of eigenvalues, variance contribution rate, and cumulative contribution rate.

Principal Component	Eigenvalues	Variance Contribution Rate/%	Cumulative Contribution Rate/%
First	4.0702	50.877	50.877
Second	1.4717	19.396	69.273
Third	1.0117	12.646	82.919
Fourth	0.5169	6.861	89.780
Fifth	0.4535	5.669	95.449
Sixth	0.2900	2.226	97.675
Seventh	0.1509	1.887	99.562
Eighth	0.0350	0.438	100.000

**Table 3 sensors-22-09892-t003:** The load number of each eigenvalue on the first five principal components.

Characteristic Parameters of the Gravity Field	First Component	Second Component	Third Component	Fourth Component	Fifth Component
σ	0.94166	−0.04244	−0.05438	−0.10338	0.17976
$\sigma_{S}$	0.89217	0.14041	0.07922	0.03174	0.26915
R	0.94146	−0.01915	0.05935	−0.05234	0.22774
H	−0.23897	0.83033	0.09734	0.46948	0.13206
RR	−0.80304	−0.09542	−0.23105	−0.08109	0.45517
AG	0.32659	−0.77711	−0.09100	0.52420	−0.00958
FD	−0.81539	−0.34302	0.05911	0.01189	0.26346
PK	−0.16612	−0.17225	0.96141	−0.02337	0.05284

**Table 4 sensors-22-09892-t004:** Table of Confusion Matrix.

		**Prediction**	
**Reference**		Positive	Negative
	Positive	Ture Positive	False Negative
	Negative	False Positive	Ture Negative

**Table 5 sensors-22-09892-t005:** Statistical comparison of test results.

Test	PT/%	PF/%	A/%	AA/%
1	92.00	83.19	87.00	87.60
2	87.50	93.97	91.25	90.74
3	85.71	99.14	93.50	92.43
4	92.86	99.57	96.75	96.22

**Table 6 sensors-22-09892-t006:** Statistical result of simulation and classification.

Region	Matching Probability/%	Category Labels
Region-A	92.25	suitable
Region-B	89.50	suitable
Region-C	47.06	unsuitable
Region-D	65.31	unsuitable

**Table 7 sensors-22-09892-t007:** Matching error statistics (in n mile).

Region	Direction	Max	Min	Mean	Std
	Latitude	0.0569	0.0563	0.0566	0.0002
Region-A	Longitude	0.14547	0.0004	0.0607	0.0422
	Radial	0.1560	0.0567	0.0881	0.0303
	Latitude	0.2183	0.0005	0.0932	0.0598
Region-B	Longitude	0.0217	0.0201	0.0209	0.0005
	Radial	0.2190	0.0211	0.0972	0.0567
	Latitude	11.708	0.0157	4.9348	3.4992
Region-C	Longitude	6.1323	2.7586	4.4446	0.9890
	Radial	13.217	3.4531	6.8902	3.1339
	Latitude	1.5691	0.5825	1.0758	0.2891
Region-D	Longitude	6.2376	0.0414	2.6116	1.8381
	Radial	6.4319	0.8289	2.8950	1.7479

## Data Availability

Not Applicable.

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
