# Peer review of "Classification of Gravity Matching Areas Using PSO-BP Neural Networks based on PCA and Satellite Altimetry Data over the Western Pacific"

_sensors, 2022, doi:10.3390/s22249892_

Round 1

Reviewer 1 Report

Overall the methods appear appropriate for the study, which uses open data sets and applies a series of metrics for features and feasibility. This study should provide a good baseline for future work to utilize to continue efforts in this area.

Author Response

We would like to thank you for giving us the opportunity to revise our manuscript, " Classification of Gravity Matching Areas Using PSO-BP Neural Networks based on PCA and Satellite Altimetry Data over the Western Pacific " and also thank the reviewers for their careful reading and thoughtful comments on the previous draft. These comments are valuable and very helpful for improving our manuscript. In the revised version, we have prepared point-by-point responses to these comments and we used the "track changes" function in the revised manuscript to mark the major changes made. We hope that the modifications we’ve made resolve all your concerns about the article.

Sincerely,

Jingwen Zong

Email: zjw19950613@163.com

There is the following question regarding the manuscript.

Overall the methods appear appropriate for the study, which uses open data sets and applies a series of metrics for features and feasibility. This study should provide a good baseline for future work to utilize to continue efforts in this area.

Response:

Thank you for your nice comment. Thank you very much for your approval of this paper, and we will study this method in more depth in the future.

Reviewer 2 Report

This study proposed a method based on PSO-BP Neural Networks to classify the gravity matching areas. The method is innovative and proved by their numerical tests. I recommend publication of the manuscript after necessary modifications.

First, the English writing should be improved. For example,

Line 141, Gravity matching navigation can work effectively is based on the gravity character grammatical error exist in this sentence;

Line 416, figures 3? ï¼›

Line 495-496,“ have changed->change?

Line 571, suitability -> suitable.

Lines 686-687, We thanks for , thanks?

I suggest the authors check the English carefully, and try to avoid grammatical errors.

Second, the literature list should be modified.

Some literature should be added, and some of the cited literature should be removed. For example,

(1) The related references on Equations (3~12) and Equations (21~24) should be cited.

(2) Line 285, the related reference should be cited.

(3) Line 587, According to other studies, which study? The references should be cited here.

(4) Lines 46, Zheng et al., 2008, 2012 should be removed, because these references are not related with the topic of this study.

I suggest the authors check the citations, and add the related references.

Third, some of the descriptions should be modified and more explanations should be added. For example,

(1) Line 484, combination matrix, it seems Equation (20) is not a matrix.

(2) What is the input data in Section 4.2?

(3) Lines 536~537, more explanation should be added.  For the readers not in this field, it is difficult to understand this conclusion.

Author Response

We would like to thank you for giving us the opportunity to revise our manuscript, " Classification of Gravity Matching Areas Using PSO-BP Neural Networks based on PCA and Satellite Altimetry Data over the Western Pacific " and also thank the reviewers for their careful reading and thoughtful comments on the previous draft. These comments are valuable and very helpful for improving our manuscript. In the revised version, we have prepared point-by-point responses to these comments and we used the "track changes" function in the revised manuscript to mark the major changes made. We hope that the modifications we’ve made resolve all your concerns about the article.

Sincerely,

Jingwen Zong

Email: zjw19950613@163.com

There is the following question regarding the manuscript.

(1) Line 141, “Gravity matching navigation can work effectively is based on the gravity character” grammatical error exist in this sentence;

Response:

Thank you for your comment.

Line 141: Significant changes in the gravity characteristic of the matching region ensure that gravity matching navigation works effectively.

(2) Line 416, “figures 3”? ï¼›

Response:

Thank you for your comment. We have corrected the relevant word in paper.

Line 416: Figures 3. Schematic view of the reference map in the China Western Pacific region (https://www.gebco.net/, June. 2022).

(3) Line 495-496,“ have changed”->change?

Response:

Thank you for your comment. We have corrected the relevant word in paper.

Line 495: Figure 6(a), the gravity characteristics of the northwest and southeast regions change significantly.

(4) Line 571, “suitability ”-> suitable.

Response:

Thank you for your comment. We have corrected the relevant word in paper.

Line 571: The suitable regions are mainly concentrated in the northwest and southeast, which is consistent with our previous prediction.

(5) Lines 686-687, “We thanks for ”, thanks?

Response:

Thank you for your comment. We have corrected the relevant word in paper.

Line 686: Thanks for the University of California-San Diego, who provided the gravity anomaly data in the Western Pacific (http://topex.ucsd.edu/, 4 August 2021). Thanks for the GEBCO provided the satellite imagery, (https://www.gebco.net/, June. 2022).

(6) Some literature should be added, and some of the cited literature should be removed. For example,â‘  The related references on Equations (3~12) and Equations (21~24) should be cited.â‘¡ Line 285, the related reference should be cited.â‘¢ Line 587, “According to other studies”, which study? The references should be cited here.â‘£Lines 46, “Zheng et al., 2008, 2012” should be removed, because these references are not related with the topic of this study.

I suggest the authors check the citations, and add the related references.

Response:

Thank you for your comment. We have added and removed some of literature in paper. The details of the changes are as follows:

â‘  About the equation 3-5 and 7-9, we added the literature “Ma, Y. Y.; Ouyang, Y. Z.; Huang, M. T, et al. Selection method for gravity-field matchable area based on information entropy of characteristic parameters. Journal of Chinese Inertial Technology(in Chinese). 2016, 24(6): 763-768”.

About the equation 6, we added the literature “Majhi, B.R. Gravitational anomalies and entropy. Gen Relativ Gravit 45, 345–357 (2013). https://doi.org/10.1007/s10714-012-1474-9”.

About the equation 21-24, we added the literature “Deepak Gupta, Utku Kose, Ashish Khanna, Valentina Emilia Balas. Deep Learning for Medical Applications with Unique Data, Academic Press,2022, Pages 31-51, ISBN 9780128241455, https://doi.org/10.1016/B978-0-12-824145-5.00005-8”.

â‘¡ Line 285, we added the literature “Kennedy J . Particle swarm optimization[J]. Proc. of 1995 IEEE Int. Conf. Neural Networks, (Perth, Australia), Nov. 27-Dec. 2011, 4(8):1942-1948 vol.4”.

â‘¢ Line 587, we added the literature “Wu L, Wang H, Chai H, Zhang L, Hsu H, Wang Y. Performance Evaluation and Analysis for Gravity Matching Aided Navigation. Sensors (Basel). 2017 Apr 5;17(4):769. doi: 10.3390/s17040769. PMID: 28379178; PMCID: PMC5422042.”

â‘£ Line 46, we have deleted the “Zheng et al., 2008, 2012”

(7)  Line 484, “combination matrix”, it seems Equation (20) is not a matrix.

Response:

Thank you for your comment. Sorry, we have a description error in the line 484.

Line 484: The principal component linear equation can be calculated as:

(8) What is the input data in Section 4.2?

Response:

Thank you for your comment. We should add some of the explanations about the input data in Section 4.2. We explained the input of the neural network in the Line344 “This paper selects the first k principal components  as input for the neural network.”

Line 502: The data in equation(20) which are processed by the PCA are as input data of the neural network, a PPBA method based on the PSO-optimized BP neural network is proposed to perform the classification and prediction of the matching region.  

(9) Lines 536~537, more explanation should be added.  For the readers not in this field, it is difficult to understand this conclusion.

Response:

Thank you for your comment. We should add some of the explanations about the table 6 in Line 536-537.

Line 536: As shown in Table 6, the results of matching probabilities in Region-A is 92.25% and in Region-B is 89.50%, both of them are greater than 85% which we set the probability of success for matching navigation. The results of classification by PPBA for both Region-A and Region-B are “suitable”, which are in good agreement with the result of matching probabilities. At the same time, the results of matching probabilities for both Region-C and Region-D are less than 85%, which are in good agreement with the "unsuitable" given by the classification result of PPBA. The above analysis shows that the PPBA method is available.

Reviewer 3 Report

The authors proposed the article titled “Classification of Gravity Matching Areas Using PSO-BP Neural Networks based on PCA and Satellite Altimetry Data over the 3 Western Pacific".  The following comments should be incorporated into the manuscript:

1. I noticed that the authors did not cite their work using the MDPI format, but rather in the text. It must be formatted in accordance with the MDPI rule.

2. The text's use of the English language should be improved.

3. The manuscript needs to include abbreviations.

4. The manuscript should emphasize novelty.

5. Graphical representation of your investigation should be there in the introduction

Section.

6. Latest relevant literature should be incorporated into the Manuscript.

7. I have seen of low qualities figs in the manuscript; therefore, all figs should be rewritten.

8. The comparison graph needs to be added to the results section.

9. How can you validate your contributed work with existing work?
